# Root-Zone Restriction Regulates Soil Factors and Bacterial Community Assembly of Grapevine

**DOI:** 10.3390/ijms232415628

**Published:** 2022-12-09

**Authors:** Muhammad Salman Zahid, Muzammil Hussain, Yue Song, Jiajia Li, Dinghan Guo, Xiangyi Li, Shiren Song, Lei Wang, Wenping Xu, Shiping Wang

**Affiliations:** 1Department of Plant Science, School of Agriculture and Biology, Shanghai Jiao Tong University, Shanghai 200240, China; 2State Key Laboratory of Mycology, Institute of Microbiology, Chinese Academy of Sciences, Beijing 100045, China

**Keywords:** grapevine, root-zone restriction, stress cultivation, bacterial community structuring, soil interactions, bacterial networking

## Abstract

Root-zone restriction induces physiological stress on roots, thus limiting the vegetative and enhancing reproductive development, which promotes fruit quality and growth. Numerous bacterial-related growth-promoting, stress-mitigating, and disease-prevention activities have been described, but none in root-restricted cultivation. The study aimed to understand the activities of grapevine bacterial communities and plant-bacterial relationships to improve fruit quality. We used High-throughput sequencing, edaphic soil factors, and network analysis to explore the impact of restricted cultivation on the diversity, composition and network structure of bacterial communities of rhizosphere soil, roots, leaves, flowers and berries. The bacterial richness, diversity, and networking were indeed regulated by root-zone restriction at all phenological stages, with a peak at the veraison stage, yielding superior fruit quality compared to control plants. Moreover, it also handled the nutrient availability in treated plants, such as available nitrogen (AN) was 3.5, 5.7 and 0.9 folds scarcer at full bloom, veraison and maturity stages, respectively, compared to control plants. Biochemical indicators of the berry have proved that high-quality berry is yielded in association with the bacteria. Cyanobacteria were most abundant in the phyllosphere, Proteobacteria in the rhizosphere, and Firmicutes and Bacteroidetes in the endosphere. These bacterial phyla were most correlated and influenced by different soil factors in control and treated plants. Our findings are a comprehensive approach to the implications of root-zone restriction on the bacterial microbiota, which will assist in directing a more focused procedure to uncover the precise mechanism, which is still undiscovered.

## 1. Introduction

Grape is the world thirst most crucial crop after potato and tomato, holding the farmer’s gate value of USD 68 billion, with a production of 77.4 million tonnes (MT) from 7.1 million hectares (MH) of the area under grapevine cultivation [1]. As the world population will be 10 billion by 2057, agricultural land and cultivation resources must be sustainable for irrigation and land utilization to improve the quality of fruit as per the United Nations (UN) sustainable development goals (SDGs) 2&12. The need of the hour is to devise sustainable cultivation technologies of adequate land and water usage and ultimately yield better fruit quality. Root restriction is an innovative, stress-inducing technique in fruit tree cultivation based on fruit trees’ vegetative and reproductive growth, which can be regulated by restricting roots into a container [2,3], thus regulating the water and nutrition absorption capacity. This technique’s ideology aligns with the United Nations (UN) sustainable development goals (SDGs) number 2&12, which state zero hunger and sustainable management and efficient use of natural resources. Root restriction cultivation enables us to save resources on fertilization and irrigation. This method has been adopted in numerous vineyards in China due to its ecological and financial advantages, and it has shown promising results in the commercial production of grapes. Root restriction is a suitable method for high-density cropping. Another vital feature of root restriction cultivation is that it assisted in overcoming the low degree of sweetness in grapevines grown in southern China regions as a result of the hot, humid weather [4]. Root restriction is suitable for high-density, high-yield and high-efficiency cultivation [5]. Numerous studies on different fruit, such as grapes [2,6,7], nectarines [5], sweet cherry [8], chilli [9], apple [10] and tomato [11], have revealed the remarkable changes that can suppress vegetative growth and boosts the reproductive growth as well. The primary goal of implementing this strategy is to have the most delicate fruit quality possible, which is regulated by an increase in the size, colour, flavour, anthocyanin, and content of total soluble solids [3,4,12,13]. Grapevines grown with root-zone restriction displayed distinct growth behaviours as compared to standard cultivation. By the physiological stress applied, the primary roots were thinner and longer [14,15], while secondary and fibrous roots exhibited a high absorption capacity. The vine vitality, in particular, was severely reduced [16,17]. Meanwhile, root restriction controlled the distribution and partitioning of assimilates between vegetative and reproductive organs, resulting in better berry quality [2,4,12]. Phytohormones are also reported to be increased by root restriction, such as abscisic acid (ABA) and salicylic acid (SA) [4,18]. Root restriction causes high stress to peach tree growth as characterized by the photosynthesis efficiency, tree growth, fruit quality and yield [5]. The bacterial community structuring and diversity in root restriction is still a mystery, despite investigating these numerous physiological features.

Plants are aided by their associated microbes regarding host health, fruit quality, and adaptation to their fluctuating environmental conditions. Plants and bacteria form intricate and dynamic relationships that can vary from beneficial to commensal or harmful [19]. Their functionalities, such as soil quality, host productivity, and host health, are improved by the mineralization of soil organic matter, activation of plant defence systems even the synthesis of antibiotics against phytopathogens are all examples of direct or indirect processes [20,21,22]. Plant-associated bacteria populate as epiphytes and endophytes. The soil surrounding the plants acts as a microbial reservoir. Plant–microbe interactions are complex, with plant species and soil types influencing the soil microbial ecosystem and having a critical role in plant defence [23,24]. Bulgarelli [25] proposed that edaphic variable, substrate-driven community selection within the rhizosphere, and host genotype, shape the soil microbial architecture. Endophytic bacteria invade roots, leaves and eventually reproductive structures after rigorous filtering. It is now commonly known that the formation of a bacterial community in plants does not occur randomly but is governed by certain assembly principles [25,26,27]. Soil type is one of the elements influencing the organization of bacterial communities in plants [28,29]. Others are plant compartment [30], host genotype/species [31], plant immune system [27], plant trait variation/developmental stage [32,33], and residence time/season [34]. Although the bacterial microbiota is well studied in diverse environments and plant species, the influence of a stressful technique, root-zone restriction on the grapevine, is still unexplored. Here we hypothesized the role of the root-zone restriction technique in recruiting the plant-associated rhizospheric bacterial microbiota and its interaction with the soil edaphic factors. It increases the root formation, increasing the surface area and, consequently, critical bacterial taxa corresponding with plant organs and soil. It will assist us in comprehending the microbial terroir of grapevines grown in root-zone-restricted cultivation. Our primary focus was on how the root-zone restriction influences the bacterial community structures at different growth stages and its relationship with fruit quality.

## 2. Results

### 2.1. Biochemical Dynamics of Root Zone Restriction Approach on Grapevine Fruit

Comparing the significance of the biochemical parameters between treatment and control, other than Ph, the rest were statistically non-significant at the pre-veraison stage. TA and pH were significant, while anthocyanin and vitamin C depicted identical results at the veraison stage. All parameters were highly significant when compared between root-zone restriction and control at the maturity stage, as shown in Table 1.

### 2.2. Bacterial Taxonomic Diversity throughout the Grapevine Influenced by Root Zone Cultivation

The bacterial diversity of the Muscat Hamburg cultivar was determined in the rhizosphere soil, white roots, leaves, flower, and berries at three major phonological stages. They are full bloom, version, and maturity, comparing treatment (root-restricted plants) and control (non-root-restricted plants) by Illumina MiSeq sequencing of 16S rRNA genes.

A total of 1,600,033 V3–V4 amplicon sequence variants (ASVs) were documented from the rhizosphere, roots, leaves, flowers, and berry at full bloom, veraison, and maturity. The average length of sequences was 376 bp, while the majority were 407 bp, and samples were rarefied to 66,983 sequences each. Overall, 13,541 ASVs were identified after quality filtering from 72 samples (24 samples were run in 3 replicates) and approximately 600 ASVs per sample (Appendix A Appendix A). Rhizosphere soil of control plants has 1982, while treated plants have 1770 ASVs. White roots had 927 and 1026 ASVs of control and treated plants, respectively. Leaves of control and treated plants have 1805 and 1997, respectively. Flower resided 893 and 589 ASVs in control and treated plants, respectively. Berry of control plants contained 1456, while 1085 were in treated plants. Sequencing statistics per sample and the number of observed ASVs (Appendix A Appendix A). The findings of a comparison of rarefaction curves (Appendix A Appendix A) as a function of sampling depth and observed outs revealed that all curves are close to saturation. As a result, the richness of the samples has been meticulously documented or sequenced [35,36].

### 2.3. Impact on Bacterial Community Structure in Plant Organs across Three Growth Stages

Microbial community structure across the grapevine habitats is exposed at the phylum level (Figure 1). The relative abundance of Cyanobacteria was 40%, Proteobacteria 32%, Firmicutes 12%, and Bacteroidetes 4%, which were most abundant across all the treated and control plant organs at all growth stages. The highest relative abundance of Cyanobacteria was 94.03% in maturity leaf treatment (MLT) and the lowest in full bloom soil control (FBSC). Proteobacteria was profused in veraison soil treatment (VST) 90.42% and minimal abundant in MLT. Firmicutes were at 59.57% in the soil of control plants at the full bloom stage (FBSC), and 0.13% in the full bloom white roots treatment (FBWRT). Bacteriodiates were maximum classified at 14.04% in V_B_T (veraison, berry of treated plant) and were minimum in maturity white roots control (MWRC). Regarding the plant organs, rhizosphere soil had a maximum relative abundance of Proteobacteria with 90.42% at the veraison stage of treated plant (VST) and Cyanobacteria with 0.25% at the full bloom stage of control plants (FBSC). White roots were composed of 84.87% of Cyanobacteria at the maturity stage of control plants (MWRC) and a minimal abundance of Bacteroidetes with 0.12% at the maturity stage of control plants (MWRC). Leaves of the treated plant at maturity stage (MLT) had 94.03% of Cyanobacteria and 0.29% of Bacteroidetes at maturity stage (MLT). The flower of the treated plant at full bloom (FBBT) and berry of the treated plant at the veraison stage (VBT) was enriched with Proteobacteria at 59.17% and 64.46%, respectively. A minimal abundance of Firmicutes was observed in full bloom berry control (FBBC) at 4.25% and 2.66% at maturity berry control (MBC). For a detailed relative abundance of these phyla, see Appendix A Appendix A.

### 2.4. Richness and Diversity of Bacterial Communities Associated with Grapevine Habitats

Alpha diversity analysis was used to investigate the species richness and evenness of a bacterial ecosystem within the rhizosphere (soil and white roots) and phyllosphere (leaves, flower and berry). To validate our sample size, species accumulation curve and rank abundance curve were formulated (Appendix A Appendix A). Alpha diversity was assessed by Chao1 and Shannon indices (Figure 2). The rhizosphere soil had Chao1 (*p* = 0.096) maximum at maturity soil treatment (MST), while Shannon (*p* = 0.0078) was at veraison soil control (VSC) (Figure 2a). White roots had Chao1 (*p* = 0.076) maximum at veraison white root control (VWRC) while Shannon (*p* = 0.21) at full bloom white root control (FBWRC) (Figure 2b). Shannon (*p* = 0.0078) and Chao1 (*p* = 0.015) were maximum at veraison leaf treatment (VLT) (Figure 2c). The flower was contemporary to the full bloom stage, so it has higher Chao1(*p* = 0.02) and Shannon (*p* = 0.031) at full bloom berry control (FBBC). Berries have higher Chao1 (*p* = 0.02) and Shannon (*p* = 0.031) at veraison berry control (VBC) (Figure 2d).

Bacterial community structures were visualized by nonmetric multidimensional scaling (NMDS) based on the Bray–Curtis distance. Dissimilarities between the organs at different phonological stages were measured with orientation to control and treatment. The soil at the veraison stage of treatment and control plants (FBST and FBSC) showed a significant community shift (r^2^ = 0.87, *p* < 0.001, stress = 0.000862). In contrast, as indicated in the plot, all other groupings were centred on a single spot (Figure 3a). In Figure 3b, white roots were most distinct between treatment and control plants at full bloom, followed by maturity (r^2^ = 0.58, *p <* 0.001, stress = 0.0917). In Figure 3c, our treatment impacted the bacterial community structure only during the full bloom and veraison stages in the leaves of treated and control plants (r^2^ = 0.68, *p <* 0.001, stress = 0.0488). In Figure 3d, the root-zone restriction influenced the community composition in the berries at full bloom and veraison, although the communities at maturity were identical (r^2^ = 0.42, *p <* 0.005, stress = 0.0771). To comprehend the beta diversity of microbial communities established as a result of the NGS, we constructed a hierarchical clustering analysis known as UPGMA. Plant tissues were similar intra-group of treatment and control but at a distance in the case of the intergroup (Appendix A Appendix A).

According to our hypothesis that microbes have a beneficial effect on fruit quality, we tried to silhouette a relation between the fruit quality parameters and the bacterial communities in the berries of treated and control plants at the veraison and maturity stage. Multivariate analysis as canonical correspondence analysis (CCA) revealed that Vitamin C (r^2^ = 0.81, *p* < 0.005), pH (r^2^ = 0.68, *p* < 0.005), and anthocyanin (r^2^ = 0.39, *p* < 0.005) were significantly correlated with the microbial communities of the berries at maturity stage in our treated plants. The variables along the CCA1 explained 18.2% of the variation, while 11.41% was along the CCA2 (Appendix A Appendix A).

### 2.5. The Preponderant Influence of Soil Parameters on Bacterial Community Structuring

Soil chemical properties are in Table 2 and Appendix A Appendix A. The relation of the soil chemical properties towards the structuring of the bacterial community was explored by constituting a redundancy analysis, which indicated that the soil nutrition, growth stage and root-zone restriction technique are the primary driver of the bacterial diversity in the soil. The two axes of the RDA plot explain 81.14% of the total variation caused by the soil factors with the Monte Carlo permutation *p* < 0.001 and the corresponding correlation factor r^2^ (Figure 4). The soil pH (r^2^ = 0.48) significantly affected the abundance of unclassified *Enterobacteriaceae*, whereas cec (r^2^ = 0.31) provoked the abundance of *Enterobacter* and *Pseudomonas* in VST. Zinc (r^2^ =0.62) turned out to be the major contributor to the diversity of Bacillus *sp* at MSC and MST. OM (r^2^ = 0.47) considerably regulated the bacterial community at FBST, and the most affected genus was *Allorhizobium-Neorhizobium-Pararhizobium-Rhizobium*. AN (r^2^ =0.61), TN (r^2^ =0.38), AK (r^2^ = 0.59), TK (r^2^ = 0.0.41), AP (r^2^ = 0.34), TP (r^2^ = 0.24), Mg (r^2^ = 0.49), and Mn (r^2^ = 0.0.34) were responsible while Fe (r^2^ = 0.73) being most correlated with *Sphingomonas* at VSC in terms of shaping the bacterial population and *Devosia, Pseudolabrys* and *Duganella* were the most affected genera.

The network analysis method was employed to investigate the connection between bacterial genera and the edaphic soil factors, along with the influence of the nutritional profile of the soil affecting the bacterial population density. The structuring was significantly different in both the control and treated plants. A correlation was also noticed between the soil edaphic factors, corresponding to their abundance and jointly venturing towards bacterial networking.

According to our sieving conditions of the constituted network r > 0.6 or r < −0.6 and *p* < 0.05, we observed 1072 (179 − ve, 893 + ve) effective interactions in control plants while 994 (463 − ve, 531 + ve) interactions were recorded in the treated plants. The analysis was performed on the genus (Appendix A Appendix A) and phylum levels. The co-occurrence network was compartmentalized into clusters connecting the edaphic factors as the bacterial niches. According to the network diagram of control plants (Figure 5a), it was observed that there is a correlation among the soil elements, and they are mutually steering the community configuration. The correlation between AN and AK was r^2^ = 0.9; *p* > 0.0009, Mg and AN were r^2^ = 0.93; *p* > 0.0002, while Mg and AK were r^2^ = 0.83; *p* > 0.005. The maximum number of nodes were related to Firmicutes, Proteobacteria, Acidobacteria, Actinobacteria and Bacteroidetes. TK was reported to influence the genera under the Bacteroidetes and Dababacteria significantly. TP, Mn and Zn were correlated with Cyanobacteria. CEC, OM and AP governed Chloroflexi and Planctomycetes.

The relationship between the treatment and soil factors was different, as observed in the control plants. More nodes were connected and concentrated between TK, Mg, AP and Fe. According to Figure 5b, there was a significant negative correlation between AP and TK r^2^= −0.8; *p* > 0.004, Mg and AP r^2^= 0.866; *p* > 0.002 while Mg and TK were r^2^= −0.82; *p* > 0.005. Proteobacteria, Acidobacteria, Chloroflexi, Firmicutes, Bacteroidetes, Gemmatimonadetes, Deinococcus-Thermus and Rokubacteria were majorly influenced by TK, Mg and AP, respectively. Fe also retained a considerable number of nodes after the above-stated three elements. A highly positive correlation was identified between the genera belonging to Deinococcus-Thermus, Verrucomicrobia, Dadabacteria, BRC1, Hydrogenedentes, WS2 and Fe in the soil of treated plants.

### 2.6. Species Difference Analysis and Notable Species

We used the Venn diagram for community analysis to observe the core and unique ASVs among different plant organs. 155 core ASVs were shared between the soils (Appendix A Appendix A), most belonging to phylum Firmicutes (65%). Thirty-five were in white roots and seventy-eight in the leaves from Cyanobacteria (Appendix A Appendix A), with a relative abundance of 85% and 88%, respectively. While 157 were in the berries Proteobacteria (68%) (Appendix A Appendix A) of different stages independent of the treatment and growth.

The LEfSe (linear discriminant analysis effect size) analysis may directly perform difference analysis on all classification levels. At the same time, it emphasizes the discovery of distinct species differences between groups, i.e., biomarkers. The histogram of LDA values distribution shows the significantly different taxon. An LDA score of a minimum of 2 was used to identify diverse bacterial assemblages with statistical significance (*p* < 0.05). Our treatment (root restriction) showed more significant bacterial groups than control plants. Most V_S_T contains class Gammaproteobacteria, followed by an abundance of phylum Bacteroidetes in V_B_T. FB_B_T had class Clostridia in enormous numbers. Last but not least, M_L_T had mainly recruited the class Alphaproteobacteria (Appendix A Appendix A).

### 2.7. Metagenome Prediction

Using the bacterial 16S rRNA gene sequence and PICRUSt algorithm, bacterial community metabolic functions were estimated. The KEGG pathway hierarchy level 2 classified 45 metabolic pathway subfunctions into six groups. According to Appendix A Appendix A, only the top 10 most abundant pathways are displayed. Carbohydrate metabolism was higher in soil and white roots, followed by amino acid and energy metabolism. The predicted microbial metabolism in leaves was the same as in white roots but with increased abundance. More metabolic pathways were predicted in the berries of treated plants, majorly at the maturity stage. Overall, three predicted metabolic pathways surpassed all the predicted ones, such as carbohydrate, amino acid and energy metabolisms.

### 2.8. Microbial Co-Relation Network Analysis

In all networks, we found a considerably greater amount of co-occurrence interactions as compared to mutual exclusions. Microbial associations of the rhizosphere soil of the control plant and treated plant show us the correlation between them. According to the Appendix A Appendix A) network association among our samples, the soil of control plants had 426 nodes (vertices) corresponding to 4603 edges. The treatment plant had 436 and 4333 nodes and edges, respectively (Appendix A Appendix A). Degree assortativity (Pearson correlation coefficient) for samples was 0.535 and 0.523 for treatment control plants, respectively. The ratio of edges per node was higher in the treated plants. The relation of the bacterial phyla is depicted in (Figure 6a). The top abundant phyla are, Proteobacteria, Firmicutes, Acidobacteria and Bacteroides. Their relative abundance percentage is mentioned above in the bacterial phyla distribution topic in the results portion; an important aspect is that our treatment surpasses the control in the richness of the top bacterial phyla. Networking topologies of the soil samples for phyla are 2007 edges and 242 nodes, diameter of (longest distance) 9.26 nodes. The average path length (length between two nodes) was 3.5, degree assortativity was 0.293, transitivity (clustering coefficient, which states how deeply nodes are embedded in their surroundings, and hence the degree to which they prefer to cluster together) was 0.66 and 0.29 was the modularity index. In white roots, 209 nodes and 494 edges existed in the control plants, while 194 nodes with 310 edges were present in the treated plants. The degree of assortativity was 0.84 and 0.94 for treated and control plants, respectively. Control plants tend to have a higher ratio of edges per node in the associated network of samples (Appendix A Appendix A). Cyanobacteria, Proteobacteria, Bacteroidetes and Firmicutes were the highest abundant phyla in microbial phyla networking (Figure 6b). Other than Proteobacteria, all three were more profused in control plants. Topologies demonstrate that 271 nodes with 1137 edges, the diameter was 11.82 nodes, 4.96 was the average path length, degree assortativity was 0.425, transitivity was 0.525 and 0.64 was the modularity index. Apropos the network association of sample groups (Appendix A Appendix A), leaves were expressive to contain 323 nodes and 1216 edges of control plants, contrasting to 325 nodes and 1316 edges of treated plants. The degree of assortativity was 1.05 and 1.15 for treated and control plants, respectively. A higher ratio of edges per node in the associated network of samples existed in treated plants. Cyanobacteria, Proteobacteria, Acidobacteria, and Chloroflexi, stood maximum abundant in the treated plants rendering microbial associations (Figure 6c). Two hundred and thirty-four nodes with nine hundred and fifty-one edges, the diameter was 8.59, the average path length stayed at 3.9, Degree assortativity was recorded at 0.458, Transitivity was 0.513, and 0.516 was the modularity index. The last plant organ under our contemplation is the flower/berry. The flower of the control plants had more nodes and edges than the treated plants. A total of 358 nodes correspond to 1124 edges, while 262 nodes with 554 edges in control and treated plants were present in the association between the samples, respectively. The degree of assortativity was 1.41 and 1.43 for treated and control plants, respectively. A higher ratio of edges per node in the associated network of samples existed in control plants (Appendix A Appendix A). Conferring to (Figure 6d) the associations between the microbial taxa at different growth stages observed in the flower/berries of control and treated plants, Proteobacteria, Cyanobacteria, Bacteriodates and Firmicutes all were responsible for fabricating the microbial architecture of the treated plants. Topologies depict that 247 nodes were consequential to 933 edges, average path length stayed at 3.9, degree assortativity was noted at 0.518, transitivity and modularity were 0.517 and 0.561, respectively. Each node’s Zi and Pi values were computed by using the modular analysis results to make a scatter plot. The Zi value represents within-module connection, whereas the Pi value represents among-module connectivity. In this network, the majority of nodes were in the peripherals (regarded as habitat specialists), three module hubs (generalists), and two were on the brink between peripherals and connectors (generalists). In contrast, no node was recorded in the network hub (super generalists) in the bacterial community network (Appendix A). These structural properties enable instant and easy comparisons across large datasets from various ecosystem types in order to understand how the basic aspects of a certain habitat type may influence the assembly of microbial communities. The most connected node taxonomy changed considerably across organs of treated and control plants. A dominant specie network with a grouping abundance pie chart was also compiled to figure out the species most abundant in each plant organ, with special emphasis on growth stage and treatment (Appendix A Appendix A).

## 3. Discussion

Rhizosphere soil and plant parts uphold distinctive microbial communities, with sample type accounting for the most significant variance in microbial community structure of all parameters [37,38]. Numerous studies have reported their functionalities, mainly plant growth protentional, superior fruit quality and disease suppression [24,39,40,41]. The possible effects of plant phenological stages, composition of soil and above-ground plant organ microbial communities have been widely studied, primarily in annual crops and grapevines [20,42,43,44,45,46]. Nevertheless, no research has been reported before on the behaviour of grapevine-associated microbiota and the effect of soil properties on them under the stressful cultivation root-zone restriction, compared to control plants at three main phonological stages. Biotic and abiotic factors are an administrator of endophytic communities. The plant, including the host genotype and developmental stage, and the environment from which endophytes originated, plays a critical role in regulating plant function [47]. Our results provided a novel approach that the bacterial communities associated with plant organs considerably contrasted in richness and diversity, fruit quality is undoubtedly regulated by microbes, and soil properties have a distinct correlation with particular microbes under a stressful cultivation name root-zone restriction.

### 3.1. Root-Zone Restriction Influences the Bacterial Structuring across Successional Stages

Significant diversity was recorded among the soil samples of treatment and control plants in the veraison stage; the earlier was less diverse but highly rich than the latter. The most abundant phylum was Proteobacteria, with a rise of 60.5% in VST, with the subsequent class Gamma-proteobacteria presenting a 1.2-fold increase in VST as compared to VSC. The vital function associated with them is nitrogen fixation. Proteobacteria are well known for their part in the carbon metabolic cycle and the generation of secondary metabolites [48], encompassing spoilage and fermenting species [49]. The soil is predominated by proteobacteria [50], and they are also very stressed responsive in a constrained environment [26]. Less diversity in our treatment shows that root-zone restriction influences the soil diversity, and Proteobacteria was the dominant phyla among others. Typically, the rhizosphere is where root microbial diversity first emerges. It has been discovered that vineyard management approaches, local bio-geographical factors, and soil physicochemical properties all impact the make-up of root-associated populations [50,51,52]. The root exudates draw microorganisms from the soil around them and are carried further throughout the growing season [53]. Additionally, bacteria may develop an endophytic phenotype and spread from the inside of the roots to other plant tissues [54,55]. Here we found that Cyanobacteria was most abundant in the roots, showing a 1% increasing trend from full bloom to maturity in treated plants, while the opposite was observed in control. Followed by Proteobacteria. Steroidobacter spp., which was abundant in roots of treated plants at full bloom (7.12%) and version (3.1%) stage than in soil, may be extremely important to the physiology and growth of plants. Brassinosteroids may be crucial to the growth and physiology of plants since they have been shown to influence seed germination, stem and root elongation, vascular differentiation, fruit ripening and stress tolerance [56]. We can assume that under the effective signalling between the roots of grapevine and Cyanobacteria, it was more abundant in the roots. Cyanobacterial richness and diversity are severely affected by drought conditions [57]; perhaps in our results, due to appropriate irrigation, this trend was not observed. Factors driven by changes in host-plant genotype, structure and growth stage create a more distinct endophytic microbial population comprised solely of bacteria capable of passing into root tissues through the endodermis and pericycle stably [52,58]. Nevertheless, there was no momentous diversity between the control and treatment at maturity and veraison rather than full bloom (Figure 2b and Figure 3b) and Appendix A Appendix A. We recorded a greater number of ASVs observed in the roots of the control plant than in the treated ones. This could be because root restriction alters the root architecture [2,59]; by peculiar signalling, only specific microbes are recruited by roots. Root exudates are also altered depending on the plant genotype, species, age, nutrition and stress [20,60,61]. Our treatment also depicts the specificity of microbes at particular stages in the roots by oozing specific types of root exudates, as reported by [62,63]. Endophytes observed in the leaves were more diverse at the veraison stage (Figure 2c). Other than Cyanobacteria and Proteobacteria, we recognized the abundance of Acidobacteria and Chloroflexi. Cyanobacteria were observed across all stages and were most abundant in the treated plant’s leaves at the maturity stage, with a 2.52% increased relative abundance compared to the control. However, it remained an unidentified chloroplast. *Proteobacteria* was less abundant in leaves at the veraison stage than in the soil of treated plants. It was least abundant at later stages. Acidobacteria and Chloroflexi were more abundant at earlier stages and least abundant till maturity (Figure 1). Root-zone restriction modifies the phyllosphere’s physiology and rhizosphere, thus shrinking the vegetative growth and escalating the reproductive growth [2]. This could be a reason for the high abundance of Cyanobacteria in the leaves, it migrates to the phyllosphere to support photosynthesis and to supply an adequate amount of nutrients for plants [64], and in return, we acquire a superior quality of fruit rather in a restricted environment [3]. The morphological features of leaves, their chemistry and physiology may alter depending on the grape species, as plant genetics may influence all of these aspects. This change may also result in different microbial community structure mixes [65]. Limited reporting exists about the presence of Cyanobacteria in the phyllosphere, grapes [65], Jingpai pear [66], and wheat [64]. While a majority of reporting is about the presence in the rhizosphere of Arabidopsis [32,67], potato [68] and wheat [69]. Flower and berries were abundant with Proteobacteria across all three stages, followed by peak abundance of Cyanobacteria in berries only with an 8.82% at maturity, Bacteriodates at version with 27.75%, and Firmicutes at full bloom stage with a 345% change in abundance of treated as compared to control plants. Berries were also recorded with maximum diversity at the veraison stage (Figure 2d). γ-proteobacteria, Bacteroides and Clostridia were the most abundant classes of Proteobacteria, Bacteroides and Firmicutes, respectively. Scanty bacterial numbers have been recorded from flowers and fruits belonging to these previously mentioned phyla [54]. Several documents reveal that different microbes associated with Proteobacteria are in charge of disease resistance, increasing plant biomass, nitrogen fixation, and growth enhancement [63,70,71]. It is relevant to have its share in wine making [72]. Allied microorganisms of plant and host collaborate vigorously, thus describing a stable holobiont where the partners cooperate to improve the overall holobiont’s fitness [73]. The functionality of the microbiome is not equal to the sum of its parts. Microbes form complex networks by mutual interaction, significantly influencing ecological processes and host adaptations [74]. Variability in bacterial community structure, particularly in the rhizosphere and phyllosphere, can be attributed to uneven colonization of the root system, variation in plant physiology (root-zone restriction), the development stage, root exudates and even miscellaneous occurrences. Microbial endophytes colonizing the plant roots is a well-known topic. The soil of the control plant was more diversified than the soil of the treated plant in the current investigation, where endophytes were studied. It confirmed most ASVs found in above and lower ground samples, indicating that soil is the main reservoir for plant-associated microorganisms, and root-zone restriction influences the recruitment.

### 3.2. Improved Fruit Quality

Regardless of this root-zone restriction that recruits particular microbes and shows less diversity, we always get superior quality yield, which instigates us to ponder on the microbe-mediated phytohormone production in a restricted or stressful environment. According to several studies, microbes generate minute levels of plant hormones to support plant growth and stress tolerance in various stressful environments, including salt, heat, drought, and metal toxicity [75,76,77]. According to Appendix A Appendix A, we recurrently have a higher quantity of TSS in the berry of root-restricted grapevines [12]. Moreover, this could also be due to the bacterial strains most abundant in the berries at the maturity stage, *B. subtilis* and *Arthrobacter* sp, as described by [78], that the total soluble sugars, wheat biomass, and salt content all were increased. ABA (abscisic acid) synthesis *by B. licheniformis* and *P. fluorescens* boosts grapevine development during water stress, as observed by [79]. Hence, the statistical relation confirms this association between microbes and bacterial abundance, yielding better fruit quality.

### 3.3. The Bacterial Population at Predisposal of Soil Elements in Root-Restricted Cultivation

Microbial niches originate from the proximity of the primary anchoring organ named roots, translocated throughout the plant and perform multiple functionalities related to plant development, fruit quality and disease resistance [54,80,81,82,83]. Soil microbial diversity is theorized to be driven by numerous biotic and abiotic factors [84], such as climate, soil properties, topography and land usage [85,86,87]. Thus, we have hypothesized that soil properties might get altered, consequently influencing the microbial community structuring under the stressful cultivation called root-zone restriction. Meanwhile, root restriction modifies plant physiology, nutrient uptake and other numerous functionalities. The soil edaphic factors in that restricted container must influence the microbial diversity, which was also compared to a control plant. Following the first section of our hypothesis, most edaphic soil factors varied significantly concerning the cultivation technique and the growth stages.

Another excellent point is that according to the Tukey’s-LSD pairwise mean comparison, the TN curtailed homogenous groupings (non-significant), while AN possessed heterogenous groupings (significant) at the full bloom and maturity stage in both the cultivations. In the value-wise comparison from (Table 2), it is evident that the amount of available nitrogen stood pretty low in the treated plant, which is aligned with our previous studies; root restriction limits the nitrogen availability in the grapevines [16,17]. Our finding also contrasts with [88], which state that nitrogen addition improved the relative abundance of Firmicutes, Actinobacteria and Acidobacteria across the N-addition gradient. However, the root-zone restriction limits the nitrogen availability, so the relative abundance of the prior mentioned bacteria possesses a diminishing trend, according to (Figure 1). Rda in (Figure 4) showed a similar trend that the effect of AN and TN on the bacterial community was not on the treated plants; instead, very low was recorded in the case of control plants. This coincides with the findings of (Liu et al., 2020) that TP, NH_4_^+^-N, and NO_3_^−^-N effects on the distribution of bacterial phyla were insignificant. Zn was positively and significantly correlated with the *Bacillus* spp., majorly in the soil at MSC and MST. This could be because Zn uptake is conducted by a rhizophagy cycle and could be affected by the atrocities caused by the root-zone restriction to the soil texture or structure. Microbes have siderophores and other methods for efficiently sequestering micronutrients, and many are motile and move around in soils to obtain nutrients. Zinc solubilization potential in *B. altitudinis* and *B. safensis* was reported by [89]. Metals also stick to microbe cell walls because the cell walls of bacteria have a net negative charge, whereas metals have a positive charge and may be absorbed by the roots [90,91].

### 3.4. Correlation between Soil and Bacterial Niches

Root-zone restriction and control plants demonstrated a high correlation between the edaphic soil factors and the soil bacterial populations. (Figure 5a,b). The gathered results imply that the existence of a relationship between the soil factors correlates with and influences the microbial existence in the rhizosphere soil. This perspective of edaphic soil characteristics and taxonomically different microorganisms will likely improve our understanding of microbial interaction. It was reported that soil edaphic factors regulate the abundant and rare microbial community structuring in the soil [92,93]. In this study, the majority of soil microbiota were persuaded by edaphic soil factors, which strengthens our belief that connected microbiota must possess a diverse ecological role which would benefit the plant’s overall growth and development. The phyla proteobacteria (Figure 5a) were most abundant in the treated plant’s soil and were significantly negatively correlated with TK and positively correlated with AP. This could be because KSBs (potassium-solubilizing bacteria) are from phylum *Proteobacteria*, *Pseudomonas*, and *Bacillus* spp., which are also enriched in our soil of treated plants [94,95]. However, if the solubilization is increased, the total phosphorus is decreased, as they are reported for their PGP activities in wheat and maize crops [96,97]. AP was also positively with Proteobacteria and negatively correlated with Deinococcus-Thermus, denoting the fact that phosphate-solubilizing bacteria are reported from both of them, and their diversity is being affected by AP in treated plants; perhaps their PGP potential is still intact [98,99,100]. Mg positively correlates with Proteobacteria and can alter bacterial diversity.

Mg is an essential constituent of chlorophyll and acts as a micronutrient. The increased Mg addition raises microbe diversification, making the microbial community more stable, which has a substantial advantage in the fight against Huanglongbing (HLB) [101]. In (Figure 5b) the control plants, we noticed an altogether different response of the soil elements towards the bacterial structuring and also, those bacteria which were negatively correlated with these elements in treatment were showing a positive correlation in the control plants and vice versa. This might be due to regular cultivation without any external stress. The majority of nodes were concentrated in MG, AN and AK. Magnesium was positively correlated with the Dienococus-Thermus phylum in control, while it was negative in treated plants. This may be because magnesium increases the production of the hydrolytic enzyme-producing bacteria, and this activity may be hindered by root-zone restriction [102]. A similar phylum remained possessed with a positive correlation with Fe in both the cultivation techniques because Fe-rich sources affect the diversity of Dienococus-Thermus, as reported by [103]. Another excellent and multifaceted bacteria, Cyanobacteria, constitute a diverse collection of soil bacteria that promote the development of their hosts through multiple direct and indirect methods, including enhanced nutrient availability, absorption, and augmentation in the plant [104]. They stood at the predisposal of different soil elements with a significantly positive correlation in treatment and control plants. TP and Mn; TN and AK were in control and treated, respectively. Cyanobacteria can be associated with nitrogen fixation in treated plants only as it stood correlated with TN. This also seconds the finding of another reported study that 50% less application of N fertilizer, at par values of grain yield with the control that received 100% recommended dose of N fertilizer, can be possible with cyanobacterial inoculation [105]. Wheat co-culture with Cyanobacteria has been shown to increase root dry weight and chlorophyll levels [106]; however, increased chlorophyll is reported in root-zone-restricted grapevines [16]. Here lies a future expedition to carve out the affinity between the cyanobacteria and root-zone restriction.

### 3.5. Root Restriction Regulates the Bacterial Networking in the Grapevine Habitats

The plant host and its associated microbes cooperate eagerly, thus, founding a stable holobiont in which the microbial cohorts thrive together to progress the overall fitness [73]. Microorganisms do not endure in isolation but rather create intricate network associations. Such networks are essential for understanding microbiome formation and response to environmental influences [107,108,109]. Meanwhile, a complex network is established among the microbial species, which significantly influences ecological processes and host adaptations since the functional capability of a microbiome is not equivalent to the sum of its constituents [74]. The network analysis highlighted notable species that might play a vital role in community stability. Even though we know very little about many of the species in the networks, we may highlight specific interactions as proof of the concept that the networks reflect genuine microbial connections and that the root-zone restriction causes its effect on the bacterial interaction. We found out that the results at node and network level topological features were different among all the plant organs of control and treated plants. In the soil of the treated plant, a more complex network was observed as compared to the control plants since more nodes and edges and the most significant number of positive interactions were also observed in it; this could be due to the reason that under the restricted container in a stressful environment soil microbes might start acting in a complex way. More positive interactions and correlations with soil microbes ultimately yield healthy grapes.

Moreover, weak network complexity was observed compared to the control plants in other grapevine habitats, such as white roots, leaves, flowers and berries. Nevertheless, the evidence supporting that weaker associations may have a diminishing or common effect on plant growth and development is also rare. This could also be validated; the decrease in network complexity may be ascribed. Increased resource availability caused by the decline process may diminish the obligation for strong cooperative and trophic connections among the functional groups of soil biota [109,110]. Regardless of the weak complexities in treated plants, we still achieve higher yields than control plants, which can be evident in increased fruit quality, chlorophyll content, and better root structuring. However, the mechanism is still poorly understood. Keystone species vital for community stability, ecological function, and plant health can be discovered through network analysis [74]. Previous research revealed that when keystone species were eliminated, the composition and function of communities altered [111]. Such taxa have a significant role in microbial ecological functions [112]. Here we have detected that most of the keystone ASVs were from Proteobacteria, Firmicutes, Cyanobacteria and Actinobacteria, respectively, as also stated by [112,113]. All of the mentioned phyla are known to contain the genera responsible for promising growth and disease suppression effects on plants [25,48,64]. Due to their critical roles in sustaining the bacterial ecosystem and network complexity, identifying the keystone taxa is a significant step that could guide toward more targeted research in plant and soil microbiome organization and engineering [114,115].

## 4. Materials and Methods

### 4.1. Plant Material and Sample Collection

Three-year-old grapevine cultivar ‘Muscat Hamburg’ (*Vitis vinifera* L.) planted in the vineyard at the School of Agriculture and Biology of Shanghai Jiao Tong University (Shanghai, China 31°11′ N, 121°29′ E) was sampled. Each grapevine was sampled for rhizosphere soil, roots, leaves, flower, and berries, during the growing season of 2019 at full bloom (May), veraison (June), and maturity stages (August), as previously reported by [83]. Grapevine nutrition and management were managed as reported by [3,12]. All samples were immediately placed in sterile bags, maintained in liquid nitrogen during field collection, and then transferred to −80 °C until DNA isolation.

### 4.2. Biochemical Characterization of Grapevine

For titratable acidity (TA), 10 mL of supernatant was diluted up to 50 mL with distilled water and titrated against 0.1 N NaOH, using 2–3 drops of phenolphthalein as an indicator and expressed as a percentage (g tartaric acid/100 mL juice).

The anthocyanin concentration in berry skins was measured at three growth stages; Berry skin (50 mg) was crushed with liquid nitrogen, homogenized in 1% (*v*/*v*) hydrochloric acid in methanol and shaken at 4 °C overnight. Concentrations were measured by the absorbance of the extract at 530 nm with an ultraviolet-visible (UV–vis) spectrophotometer [116].

Ascorbic acid was determined on juice samples diluted 1:10 (*v*/*v*) with oxalic acid and following the indophenol dye titration method [117].

Fruit juice pH was measured with a pH meter (Mettler Toledo S400, Columbus, OH, USA). The data were evaluated as a factorial experiment with three replications using a totally randomized design. Fisher LSD was used as a post hoc test to calculate the difference between the means at *p* < 0.05 with the SPSS.

### 4.3. Soil Physiochemical Characterization

As mentioned above, rhizosphere soil sampled from the plants was air-dried, sieved to 2 mm, and ground to a fine powder by an electrical grinder. Soil pH (pH) was measured using a glass electrode (Mettler Toledo Instruments, Shanghai, China) with a soil/water ratio of 1:2.5 (*w*/*v*). Soil organic matter (OM) was measured by the reduction of potassium dichromate (K_2_Cr_2_O_7_). Cation exchange capacity (CEC) was determined using a sodium acetate solution and flame photometer at a 767 nm wavelength [118]. The Kjeldahl technique was used to estimate total nitrogen (TN); total phosphorus (TP) and total potassium (TK) was digested with HNO_3_+ HCLO_4_ and then measured using inductively coupled plasma-optical emission spectrophotometry (ICP-OES Optima6000). Soil available nitrogen (AN) was determined using acid digestion followed by the alkali distillation method [119]. The molybdenum blue method (Olsen and Sommers, 1982) was incorporated to measure available phosphorus (AP) and available potassium (AK) levels by using sodium bicarbonate and ammonium acetate and then quantified using flame photometry (Cany Precision Instrument Co., Ltd. Shanghai, China). Micronutrients (Fe, Mn, Zn) were extracted using Diethylenetriamine penta-acetic acid-calcium chloride-triethanolamine (DTPA-CaCl_2_-TEA, pH 7.3) (Lindsay and Norvell, 1978) and evaluated using an atomic absorption spectrophotometer (PerkinElmer AAnalyst 800). Two-way ANOVA measured the difference between the soil properties of control and treatment plants, and Tukey’s LSD was later performed as a post hoc (*p* < 0.05) to compute the pairwise mean comparison between the soil sampling groups by using SPSS software.

### 4.4. Genomic Extraction, 16S rDNA Library Preparation and MiSeq Sequencing

The Mag-bind Soil DNA Kit (D5635-02) (Omega Bio-Tek, Norcross, GA, USA) was used for genomic DNA extraction rendering to the manufacturer’s instructions. The quantity and quality of extracted DNA were measured by a NanoDrop ND-1000 spectrophotometer (Thermo Fisher Scientific, Waltham, MA, USA) at 260 nm/280 nm and 260 nm/230 nm and agarose gel electrophoresis, respectively. The bacterial 16S rRNA gene’s V3-V4 region was amplified by using PCR and gene-specific primers 338F (5′-*GCACCTA*ACTCCTACGGGAGGCAGCA-3′) and 806R (5′-GGACTACHVGGGTWTCTAAT-3′). Sample-specific 7 bp barcodes were incorporated into the primers for multiplex sequencing. The PCR amplicons were purified using the AxyPrep DNA Gel Extraction Kit (Axygen, San Francisco, CA, USA, AP-GX-250) and quantified on a microplate reader (BioTek, Winooski, VT, USA, FLx800) with the Quant-iT Pico Green dsDNA Assay Kit (Invitrogen, Waltham, MA, USA, P7589), pooled together in equal amounts. Pair-end 2 × 250 bp sequencing was performed using the Illumina MiSeq platform with MiSeq Reagent Kit v3 (600-cycles-PE) (Illumina, San Diego, CA, USA, MS-102-3003). Illumina MiSeq sequencing was performed at Shanghai Personal Biotechnology Co., Ltd. (Shanghai, China). Detailed PCR conditions and library protocols can be seen in Appendix A.

### 4.5. Computational, Bioinformatics and Statistical Approaches for MiSeq Analysis

Majorly the analysis was performed with QIIME 2 2019.4 [120]. Prior to statistical analysis, non-bacterial sequences of chloroplast and mitochondrial origin were removed. To summarise, raw sequence data were demultiplexed using the demux plugin, and primers were cut using the cutadapt plugin [121]. Using the DADA2 plugin, the sequences were quality filtered, denoised, combined, and chimaera eliminated [122]. The preceding procedures are examined independently for each library. All libraries, the ASVs feature sequence, and the ASV table are merged after denoising, and singletons ASVs are deleted. Non-singleton amplicon sequence variations (ASVs) were aligned using mafft [123]. The feature-classifier plugin, classify-sklearn naïve Bayes taxonomy classifier, was used to assign taxonomy to ASVs [124] against the Greengenes 16S rRNA gene database (release 13.8) with 99% OTUs reference sequences [125]. A phylogeny was constructed with fasttree2 using mafft-aligned ASVs [126].

### 4.6. Diversity and Statistical Analysis

The Shannon diversity index and the Chao1 richness estimate were used to compute the alpha diversity. A box plot was constructed as a graphical representation of the alpha indices. Kruskal–Wallis rank-sum test was applied to calculate the significant effect between the groups, and the Dunn test was used as a post hoc test to verify the significance of the difference. Beta diversity estimation was based on the Bray–Curtis distance matrix [127] to examine the community dissimilarity via non-metric multidimensional scaling (NMDS) [128] and hierarchical clustering analysis (UPGMA) by using the vegan package and Uclust function in the R package. Permutational multivariate analysis of variance (PERMANOVA) was carried out using the adonis function in the vegan package [129].

To examine compositional variations in the microbial community residing in the berries across the seasonal succession and to establish associations with root-zone restriction and fruit quality parameters, we conducted a multivariate analysis called canonical correspondence analysis (CCA), and the corresponding correlation factor was used to assign the r^2^ value by using R package ‘vegan’ (v 3.5.2). Redundancy analyses (RDA) were used to examine categorical environmental factors to predict the temporal fluctuation of microbial community structure. Monte Carlo permutation tests (*p* = 0.001) were used to evaluate significant associations between environmental factors and genera distributions. The correlation factor r^2^ was assigned by the corresponding correlation calculation method by ‘envfit’ function under the ‘vegan’ package in R (v3.5.2).

To highlight the common and unique ASVs among samples or groups, a petal map (Venn diagram) was generated using the R script (Venn Diagram package). It is predicated on the existence of ASVs in different samples/groups, independent of relative abundance. Using the default parameters for LEfSe (linear discriminant analysis effect size), the ggplot2 program was used to find differentially abundant taxa between groups [130]. It combines the nonparametric Kruskal–Wallis and Wilcoxon rank-sum tests (*p*-value 0.05, LDA > 2) with the effect size of linear discriminant analysis (LDA).

### 4.7. Prediction of Metabolic Functions

PICRUSt2 (Phylogenetic Investigation of Communities by Reconstruction of Unobserved States) is a tool that forecasts the metabolic abundance based on the 16SrRNA gene sequence data and microbial reference gene in the sample [131]. PICRUSt predicts the 16S rRNA gene using three spectrum databases: KEGG, COG (Clusters of Orthologous Groups), and Rfam. Metabolism, Genetic Information Processing, Environmental Information Processing, Cellular Processes, Organismal Systems, and Human Diseases are the six categories in which metabolic pathways are classified.

### 4.8. Bacterial Network Analysis

Statistical correlations underpin bacterial community co-occurrence patterns and molecular ecological networks. Distinct networks were constructed for ASVs of each plant organ and rhizosphere soil at three phonological stages. Before network construction, MENA was used in conjunction with random matrix theory (RMT)-based algorithms to automatically find the suitable similarity threshold or the minimal strength of the link between each pair of nodes [132]. Another network analysis was shown to compute the consequence of edaphic soil factors on the bacterial community structuring in grapevines under control and treatment plants. To measure the co-occurrence, Spearman’s correlation factor (r > 0.6 or r < −0.6) was incorporated into the abundance profile of each Asv and the soil factor, which was analysed by R. Furthermore, the significant relationship at *p* < 0.05 between soil and bacterial abundance was visualized in CYTOSCAPE (Version 3.7.2).

Indices of different nodes, modules, and interactions were used to derive network topological features, which were then shown, such as average node connectedness, average path length, diameter, cumulative degree distribution, clustering coefficient, and modularity. Properties were calculated to compare the differences between bacterial communities related to the rhizosphere soil and organs (white roots, leaves, and berries) of control and treated plants. Within each network related to specific soil and organ, to define the connection distribution of individual ASVs, the Z value (i.e., Zi: within-module connectivity) and *p*-value (Pi: among-module connectivity) was utilized [133]. These individual ASVs may be classified into four groups based on the criteria: peripherals (Zi 2.5, Pi 0.62), connections (Zi 2.5, Pi > 0.62), module hubs (Zi > 2.5, Pi 0.62), and network hubs (Zi > 2.5, Pi > 0.62) [133]. ASV abundance data in all samples of this project are used by default. ASV with less than ten total sequences and less than five samples are filtered out. The sparCC algorithm is used to construct a correlation matrix by using random matrix theory, and Igraph is used to construct the association network.

## 5. Conclusions

The root-zone restriction has ramifications for managing the bacterial consortiums during the successional developmental phases of grapevines in various habitats. Different communities were engaged by the restricted plants at the veraison stage, being the most significant in terms of richness and diversity among the control and treated plants which may point out that recruitment and selection of bacterial community are at the top. Our results also suggested that soil edaphic factors act accordingly and are delimited by restricted cultivation in terms of correlation and influencing bacterial community structuring. Relatively complex and stable networking was observed in the control plants, but restriction nullifies this concept by yielding better utilization of resources in terms of superior fruit quality. Root-zone restriction and soil factors regulated the profusion of keystone taxa. Our study provides the first example of the relationship between root-zone restriction, associated microbes, soil factors, and fruit quality. It will help us to decipher more underlying mechanisms of microbial stability, exitance and community stability in a confined ecosystem.

## Figures and Tables

**Figure 1 ijms-23-15628-f001:**
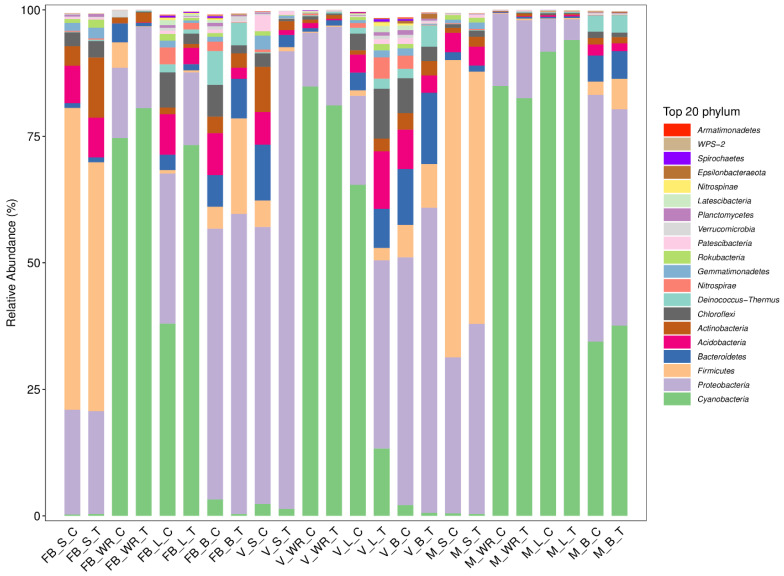
Relative abundance (%) of the significant bacterial phyla existing in rhizosphere soil (S), white roots (WR), leaves (L) and berries (B) at three phonological stages as full bloom (FB), veraison (V), and maturity (M). Root-zone-restricted plants are denoted as treatment (T) while control plants as (C).

**Figure 2 ijms-23-15628-f002:**
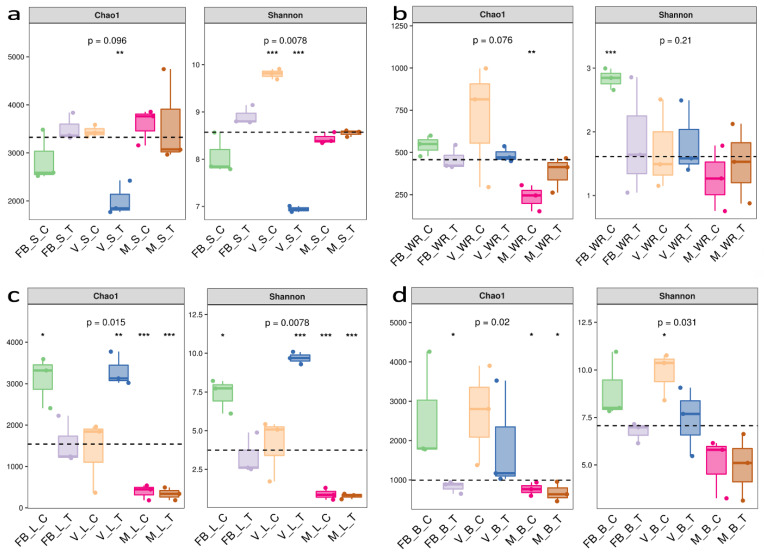
Alpha diversity indices Chao1 and Shannon. (**a**) Rhizosphere soil, (**b**) white roots, (**c**) leaves and (**d**) berries at three growth stages with comparison to treatment and control.

**Figure 3 ijms-23-15628-f003:**
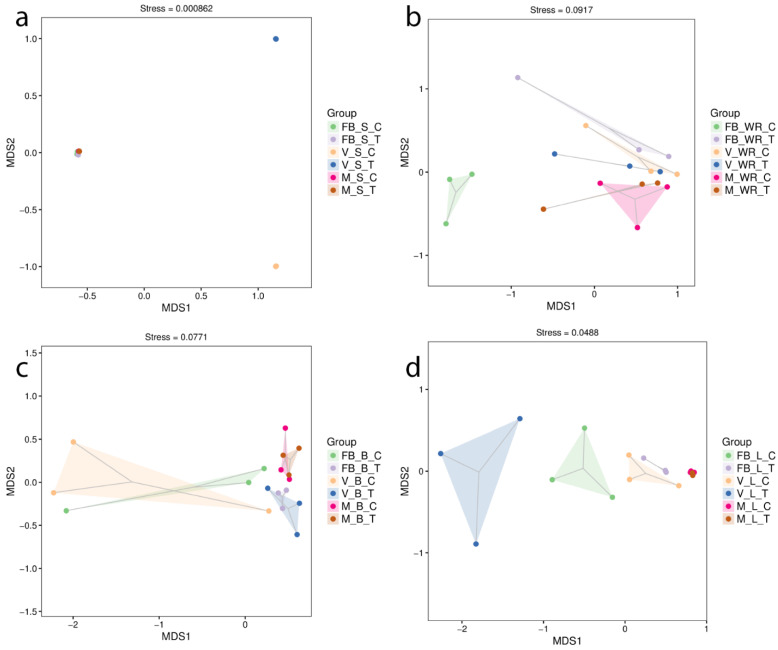
Non-metric multidimensional scaling (NMDS) for evaluation of microbial communities in samples from treated and control plants at full bloom, veraison and maturity. (**a**) Rhizospheric soil, (**b**) white roots, (**c**) leaves, and (**d**) berries.

**Figure 4 ijms-23-15628-f004:**
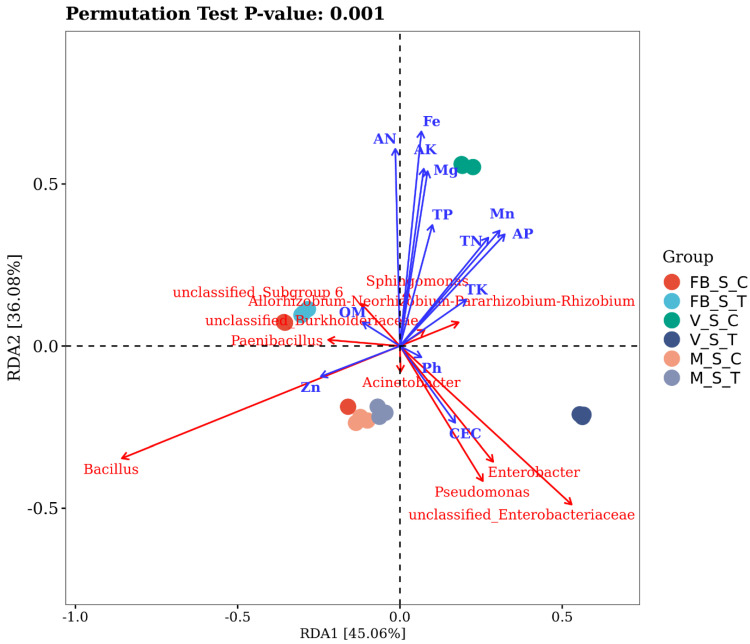
The sampling groups depicted in the colour dots, soil parameters with blue arrows and the most affected are mentioned in red colour. The smaller the *p*-value, the greater the significant impact of the overall influencing factors on the composition of the soil microbiota.

**Figure 5 ijms-23-15628-f005:**
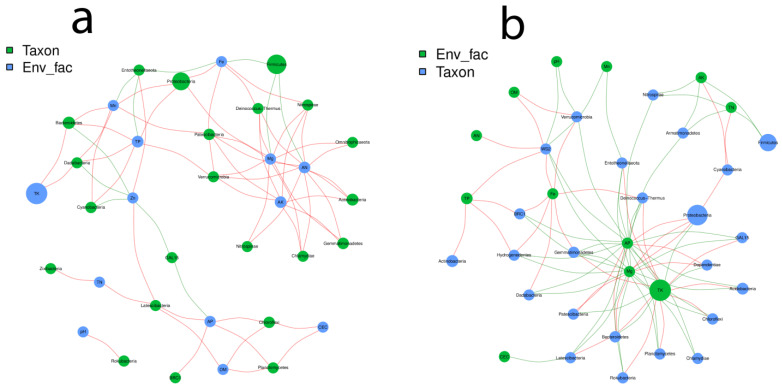
Pearson correlation between the edaphic soil factors and bacterial population in the soil. (**a**) Control plants, (**b**) treated plants.

**Figure 6 ijms-23-15628-f006:**
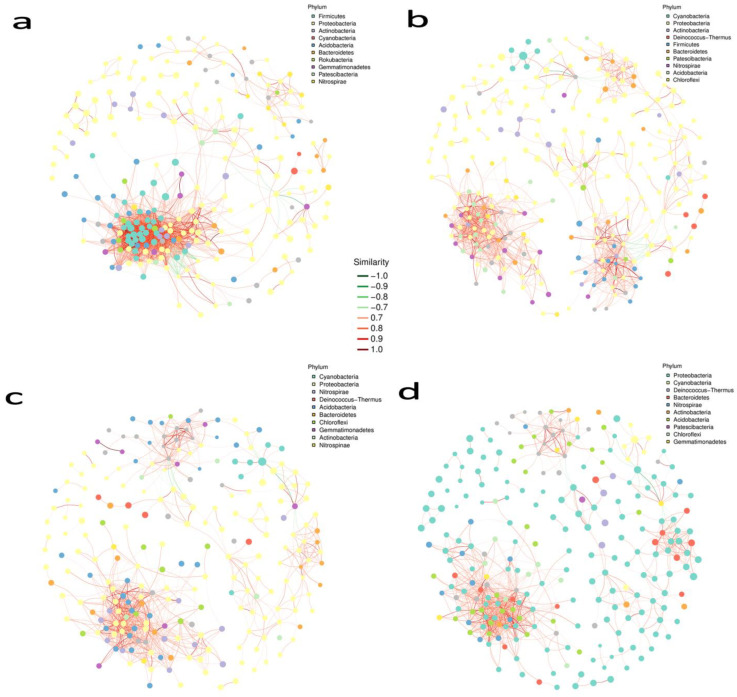
Network analysis of grape rhizosphere and phyllosphere bacterial communities. ASVs with less than 10 total sequences and less than 5 samples were filtered out. Nodes denote different interacting phyla. The top 10 modules (phyla) with the most significant number of interacting nodes are shown with their respective colours. Node size is proportional to the abundance. Edges connect different nodes with red and green lines as positive and negative correlations, respectively. (**a**) Rhizosphere soil, (**b**) white roots, (**c**) leaves, (**d**) berry.

**Table 1 ijms-23-15628-t001:** Impact of treatment and control on fruit quality parameters.

	Pre-Veraison		Veraison		Maturity	
Parameters	Control	RR	Control	RR	Control	RR
TA	4.81 ± 0.4 ^b^	5.63 ± 0.3 ^b^	2.73 ± 0.2 ^a^	1.95 ± 0.09 ^b^	1.17 ± 0.05 ^a^	0.79 ± 0.01 ^b^
Anthocyanin	0.09 ± 0.08 ^a^	0.12 ± 0.01 ^a^	3.43 ± 0.2 ^a^	3.51 ± 0.1 ^a^	99.31 ± 8.2 ^b^	117.21 ± 11.7 ^a^
pH	2.73 ± 0.1 ^a^	2.48 ± 0.2 ^b^	3.13 ± 0.01 ^b^	3.37 ± 0.05 ^a^	3.81 ± 0.07 ^b^	4.16 ± 0.03 ^a^
Vitamin C	1.42 ± 0.41 ^a^	1.47 ± 0.23 ^a^	1.77 ± 0.30 ^a^	1.80 ± 0.32 ^a^	2.02 ± 0.1 ^b^	2.21 ± 0.39 ^ab^

Mean ± standard deviation (*n* = 3) of the exact parameters followed by different letters are significantly different (*p* < 0.05).

**Table 2 ijms-23-15628-t002:** Change in soil properties at successional developmental stages of control and treated plants.

	Full Bloom		Veraison		Maturity	
	Control	Treatment	Control	Treatment	Control	Treatment
Available N	859.18 ± 55.89 ^b^	187.28 ± 9.78 ^d^	1153.49 ± 46.02 ^a^	171.29 ± 1.44 ^de^	257.84 ± 2.12 ^c^	130.40 ± 1.44 ^e^
Total N	3000 ± 0.01 ^ab^	2800 ± 0.02 ^b^	3000 ± 0.01 ^ab^	3110 ± 0.01 ^a^	2200 ± 0.01 ^c^	2100 ± 0.02 ^c^
Available P	402.16 ± 2.25 ^a^	220 ± 1.00 ^e^	397.06 ± 1.45 ^b^	352.5 ± 3.77 ^c^	214.16 ± 2.25 ^f^	271.33 ± 1.53 ^d^
Total P	611.2 ± 1.08 ^e^	1660.38 ± 142.99 ^a^	1369.8 ± 153.45 ^b^	1124.87 ± 23.07 ^c^	938.9 ± 7.98 ^d^	952.63 ± 7.46 ^d^
Available K	1063.38 ± 44.25 ^b^	433.23 ± 7.04 ^d^	1129.39 ± 17.28 ^a^	563.15 ± 9.17 ^c^	304.24 ± 2.12 ^e^	334.37 ± 2.03 ^e^
Total K	8290.13 ± 400.4 ^c^	17,308.16 ± 1371.81 ^a^	17,047.66 ± 1520.11 ^a^	14,850.33 ± 903.31 ^b^	16,221.16 ± 1511.78 ^ab^	15,971.5 ± 584.52 ^ab^
Fe	55.91 ± 2.16 ^b^	54.35 ± 1.57 ^bc^	68.70 ± 3.01 ^a^	51.37 ± 1.05 ^c^	55.58 ± 3.86 ^b^	50.62 ± 0.75 ^c^
Mn	18.08 ± 0.76 ^d^	18.57 ± 0.62 ^d^	34.06 ± 0.51 ^a^	22.06 ± 1.70 ^c^	25.96 ± 1.42 ^b^	27.83 ± 1.05 ^b^
Zn	18.83 ± 0.86 ^a^	13.26 ± 0.50 ^b^	8.40 ± 0.44 ^c^	12.77 ± 1.13 ^b^	8.67 ± 1.21 ^c^	8.71 ± 0.99 ^c^
Mg	384.79 ± 7.52 ^b^	152.03 ± 3.76 ^e^	451.32 ± 4.10 ^a^	214.26 ± 7.59 ^c^	194.86 ± 4.33 ^d^	198.89 ± 3.41 ^d^
CEC	14.00 ± 0.50 ^c^	15.73 ± 0.56 ^ab^	11.9 ± 0.2 ^d^	16.47 ± 0.54 ^a^	10.70 ± 0.37 ^e^	15.43 ± 0.56 ^b^
OM	3.93 ± 0.15 ^a^	4 ± 0.1 ^a^	3.53 ± 0.15 ^bc^	3.80 ± 0.26 ^ab^	3.23 ± 0.15 ^c^	3.58 ± 0.19 ^b^
pH	6.36 ± 0.45 ^a^	6.10 ± 0.06 ^a^	6.45 ± 0.45 ^a^	6.38 ± 0.30 ^a^	6.45 ± 0.23 ^a^	6.67 ± 0.25 ^a^

Mean ± standard deviation (*n* = 3) of the exact parameters followed by different letters are significantly different (*p* < 0.05). The unit of all elements is the same mg/kg.

## Data Availability

Data are available on the request from the corresponding author.

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
