# Peer review of "Root-Zone Restriction Regulates Soil Factors and Bacterial Community Assembly of Grapevine"

_ijms, 2022, doi:10.3390/ijms232415628_

Round 1
Reviewer 1 Report
General comments
The topic of the submitted manuscript well fits the aim and scope of International Journal of Molecular Sciences, but some minor revisions, as reported below.
A deep English grammar and language revision is strongly suggested.
Specific comments and remarks are listed below point by point.
Abstract
The Abstract section is too long. It has to be summarised in order to make it more immediate and concise, highlighting the main findings.
1. Introduction
- The following sentences “This technique's ideology aligns with the United Nations (UN) sustainable development goals (SDGs) number 2&12, which state zero hunger and sustainable management and efficient use of natural resources. Root restriction cultivation helps you save resources on fertilization and irrigation. This method has been adopted in numerous vineyards in China due to its ecological and financial advantages, and it has shown promising results in the commercial production of grapes. Root restriction is suit-able method for high density cropping. Another key feature of root restriction cultivation is that it assisted in overcoming the low degree of sweetness in grapevines grown in southern China regions as a result of the hot, humid weather.” need appropriate references.
- The following conclusion “Our findings develop the understanding of bacterial ecology and interactions in grapevines grown in root-zone restriction and provide directions for enhancing productivity.” sounds more appropriate for the Abstract or Conclusions section.
- It is recommendable to indicate at the end of the Introduction section the main employed characterisation techniques in order to achieve the indicated purpose.
4. Materials and Methods
4.1. Plant material and sample collection
- Even if reported elsewhere, more details about the sampling and the Grapevine nutrition and management have to be reported.
- The paragraph “Biochemical characterization of grapevine” has not been numerated.
Author Response
Response to Reviewer 1 Comments
We express gratitude to the reviewer for providing us with valuable feedback and suggestions for improving our manuscript titled, “Root zone restriction regulates soil factors and bacterial community configuration of Grapevine”. We believe that our manuscript is much improved as a result of the feedback. We appreciate the time and commitment involved in helping us to meet the International Journal of Molecular Science (IJMS), Section Molecular Plant Sciences and Special issue “Stress Physiology and Molecular Biology of Fruit Crops”. We hope this letter will be helpful for you to assess whether the new version of our manuscript satisfactorily addresses the reviewer's comments and concerns. We hope you agree with the actions we have taken to address your comments. If you think we failed to meet the reviewer's expectations adequately, we stand ready to respond further and would welcome the opportunity to do so. The remainder of this letter contains our point-by-point responses to the reviewer's comments.
Abstract:
Point 1: The Abstract section is too long. It has to be summarized in order to make it more immediate and concise, highlighting the main findings.
Response 1: Dear reviewer, thank you for your valuable suggestions. The abstract has been modified according to your comments.
Introduction:
Point 1: The following sentences “This technique's ideology aligns with the United Nations (UN) sustainable development goals (SDGs) number 2&12, which state zero hunger and sustainable management and efficient use of natural resources. Root restriction cultivation helps you save resources on fertilization and irrigation. This method has been adopted in numerous vineyards in China due to its ecological and financial advantages, and it has shown promising results in the commercial production of grapes. Root restriction is a suitable method for high-density cropping. Another key feature of root restriction cultivation is that it assisted in overcoming the low degree of sweetness in grapevines grown in southern China regions as a result of the hot, humid weather.” This needs appropriate references.
Response1: dear reviewer, the reference for this statement has been added in the line
Point 2: The following conclusion “Our findings develop the understanding of bacterial ecology and interactions in grapevines grown in root-zone restriction and provide directions for enhancing productivity.” sounds more appropriate for the Abstract or Conclusions section.
Response 2: Dear Reviewer, thank you very much for pointing out an important mistake. The lines pointed out were already a part of the conclusion and hence have been removed from the introduction.
Point 3: It is recommendable to indicate at the end of the Introduction section the main employed characterization techniques in order to achieve the indicated purpose.
Response 3: Dear reviewer, according to our understanding the point mentioned by you is already mentioned at the ending lines.
Material and Methods:
Point 1: Even if reported elsewhere, more details about the sampling and the Grapevine nutrition and management have to be reported.
Response 1: Dear reviewer some details have been added to the mentioned subsection. If we add all of the previous details then the material method portion will be lengthier.
Point 2: The paragraph “Biochemical characterization of grapevine” has not been numerated.
Response 2: Dear sir, the subsection named as “Biochemical characterization of grapevine” has been number 4.2

Reviewer 2 Report
Authors has taken root zone restriction approach to study the soil factors and bacterial community configuration on the grapevine production, which is a novel approach, and is a suitable method for high yield, density, and efficiency in cultivation of this crop. This approach could help to aid the microbes associated with their fruit quality, host health, and adaptation to their fluctuating environmental conditions, that may ultimately lead to increase the crop production. The article presents previously unpublished data and the overall research, and the findings given by the authors are good and appreciable. The research findings could be published in this journal after some minor corrections and modifications.
Some comments and recommendations are listed below:
Authors should improve the quality of English grammar in the text, as well as check punctuations, spaces, incomplete sentences, overlapping sentences, spellings, etc., thoroughly throughout the MS.
In Figure 5, indicators are not clearly visible, authors should provide the clear figure.
In Line 122, describe ASVs, and then use abbreviation for the same throughout the MS.
In Line 624, ‘HNO3+ HCLO4 and then measured using Inductively coupled’ should be corrected for the superscript and nomenclature of chemicals.
In Line 632, ‘DTPA-CaCl2-TEA’, should be corrected.
Figure S2 is blur, provide the clear image.
In S1 Text, abbreviation for PCR should be corrected, also there are very large overlapping in the words in sentences. So, check and correct.
Overlapping’s between words, spaces, punctuations should be checked throughout the MS.
All references should be carefully checked for spacing, italics for generic names, bold, and proper formatting as per journal guidelines.
Author Response
Response to Reviewer 2 Comments
We express gratitude to the reviewer for providing us with valuable feedback and suggestions for improving our manuscript titled, “Root zone restriction regulates soil factors and bacterial community configuration of Grapevine”. We believe that our manuscript is much improved as a result of the feedback. We appreciate the time and commitment involved in helping us to meet the International Journal of Molecular Science (IJMS), Section Molecular Plant Sciences and Special issue “Stress Physiology and Molecular Biology of Fruit Crops”. We hope this letter will be helpful for you to assess whether the new version of our manuscript satisfactorily addresses the reviewer's comments and concerns. We hope you agree with the actions we have taken to address your comments. If you think we failed to meet the reviewer's expectations adequately, we stand ready to respond further and would welcome the opportunity to do so. The remainder of this letter contains our point-by-point responses to the reviewer's comments.
Point 1: Authors should improve the quality of English grammar in the text, as well as check punctuations, spaces, incomplete sentences, overlapping sentences, spellings, etc., thoroughly throughout the MS.
Response 1: Dear reviewer, thank you for such encouraging remarks. According to your suggestion of the English language and editing, we assure you that we have counter-checked the grammatical mistakes and have improved the English quality of the original article.
Point 2: In Figure 5, indicators are not clearly visible, authors should provide a clear figure.
Response 2: Dear sir, we have tried to improve the quality of Figure 5 as mentioned by you in the manuscript.
Point3: In Line 122, describe ASVs, and then use the abbreviation for the same throughout the MS.
Response 3: dear reviewer the ASVs have been described at first in line 199 at the start of the paragraph.
Point4: In Line 624, ‘HNO3+ HCLO4 and then measured using Inductively coupled’ should be corrected for the superscript and nomenclature of chemicals.
Response4: Dear reviewer, your suggestion has been rectified in the line number 627
Point5: In Line 632, ‘DTPA-CaCl2-TEA’, should be corrected.
Response5: dear reviewer, the mentioned mistake has been corrected in the line 635
Point 6: Figure S2 is blur, provide the clear image.
Response 6: The clear image has been uploaded in the section of supplementary data.
Point7: In S1 Text, abbreviation for PCR should be corrected, also there are very large overlapping in the words in sentences. So, check and correct.
Response7: the abbreviation of PCR has been added, moreover the overlapping has been corrected.
Point8: Overlapping’s between words, spaces, punctuations should be checked throughout the MS. All references should be carefully checked for spacing, italics for generic names, bold, and proper formatting as per journal guidelines.
Response8: the overlapping between words, spaces, punctuations and grammatical errors have been checked through MS. All the references are arranged to the journal’s citation style.
